ⓐ | **Open Peer Review** | Bacteriology | Research Article

# Genomic and stress resistance characterization of *Lactiplantibacillus plantarum* GX17, a potential probiotic for animal feed applications

Yangyan Yin,[1,2,3] Chunling Li,[1,2,3] Zhe Pei,[4] Changting Li,[2,3] Zhongwei Chen,[2,3] Huili Bai,[1,2,3] Chunxia Ma,[1,2,3] Meiyi Lan,[1,2,3] Jun Li,[2,3] Yu Gong,[5] Jing Liu,[5] Ling Teng,[1,2,3] Leping Wang,[2,3] Zhongsheng Qin,[2,3] Ezhen Zhang,[6] Hao Peng[2,3]

**ABSTRACT**  Lactobacilli, recognized as beneficial bacteria within the human body, are celebrated for their multifaceted probiotic functions, including the regulation of intestinal flora, enhancement of body immunity, and promotion of nutrient absorption. This study comprehensively analyzed the genotypic and phenotypic characteristics of *Lactiplantibacillus plantarum* (*L. plantarum*) strains isolated from the intestines of healthy chicks and assessed their potential as probiotics. The assembled genome consists of 29,521,986 bp, and a total of 1,771 coding sequences (CDSs) were predicted. Based on the entire genome sequence analysis, 50 stress resistance genes and seven virulence factors were identified. The results of the phenotypic experiments showed that the strain had good resistance to high temperature, low temperature, acid, alkali, salt, artificial gastrointestinal fluid, and strong antioxidant capacity. Additionally, transcriptomic analysis confirmed that under stress conditions, the expression levels of key genes were significantly upregulated. Therefore, the phenotypic characteristics of *L. plantarum* GX17 align well with its genotypic features, demonstrating promising probiotic properties. This strain holds great potential as a probiotic candidate, and further investigation into its beneficial effects on human health is warranted.

**IMPORTANCE**  In humans, *Lactiplantibacillus plantarum* may synergize with host microbiota to ameliorate dysbiosis-related pathologies, enhance immunomodulation, and facilitate micronutrient bioavailability. For livestock, its application could improve feed conversion ratios, suppress enteric pathogens through competitive exclusion, and mitigate antibiotic overuse, "a critical strategy in One Health frameworks." Further investigations into strain-specific mechanisms (e.g., postbiotic metabolites, quorum sensing regulation) are warranted to translate these genomic-phenotypic advantages into sustainable health solutions across species.

**KEYWORDS**  *Lactiplantibacillus plantarum* GX17, genome-wide, gene prediction, stress resistance genes, stress resistance

P robiotics, serving as antibiotic alternatives, have gained widespread application in livestock and poultry farming due to their beneficial effects on disease prevention, control, and growth promotion, as reported in numerous studies (1–4). Among these, lactic acid bacteria (LAB), a class of frequently utilized probiotics, have been shown to enhance the growth performance of livestock and poultry (5, 6) and offer effective prevention and control against epidemics (7). Lactobacilli, a subgroup of LAB, are Gram-positive, catalase-negative bacteria comprising over 220 active species. Studies have indicated genetic and physiological variations among Lactobacilli from different ecological niches (8). Certain species, like *Lactobacillus texans* and *Lacticaseibacillus*

Address correspondence to Ezhen Zhang, zhang281@126.com, or Hao Peng, hpeng2006@163.com.

Yangyan Yin and Chunling Li contributed equally to this article. The order of authorship is determined based on seniority.

The authors declare no conflict of interest.

See the funding table on p. 19.

*rhamnosus*, inhabit limited ecological niches (9), while *Lactiplantibacillus plantarum* is found in diverse environments, including dairy (10, 11), vegetables (12), meat (13), silage (14), wine (15), and the gastrointestinal, vaginal, and genitourinary tracts (16–19). This wide presence underscores *L. plantarum*'s adaptability and metabolic versatility (20). Moreover, *L. plantarum* is utilized in fermenting dairy products like cheese and kefir, as well as in meat products, vegetables, and beverages (21, 22). Our laboratory identified *L. plantarum* GX17, which exhibits inhibitory effects against a range of foodborne pathogens, including *Salmonella typhimurium*, *Escherichia coli* (*E. coli*), and *Staphylococcus aureus* (*S. aureus*) (23). Following safety verification, *L. plantarum* GX17 was added to the diets of yellow-feathered broilers as a probiotic supplement, resulting in enhanced growth, improved feed conversion ratios, better intestinal health, and boosted immune function in broilers (24). Given the advantageous attributes of LAB, it's posited that its genome harbors functional genes crucial for its probiotic efficacy and environmental resilience (25–27). While current research on stress resistance genes has primarily focused on pathogenic bacteria, with less emphasis on probiotics, especially *L. plantarum*, this study aims to bridge this gap (28–30). With the rapid development of genomics and bioinformatics technology, genomic sequencing and analysis of potential probiotic strains, such as *L. plantarum*, have become extremely useful for obtaining sufficient information on safety and functional characteristics. This progress not only facilitates the understanding of their genetic background and physiological functions but also provides a scientific basis for strategies related to disease prevention and treatment (31). Therefore, this study aims to investigate the *in vitro* stress resistance of *L. plantarum* GX17 and to study its gene pool in depth by sequencing the genome of *L. plantarum* GX17 and combining genotype and phenotype, exploring its stress resistance mechanisms, and evaluating its probiotic qualities at a molecular level. The suitability of *L. plantarum* GX17 as a potential high-quality strain was explored at the molecular and phenotypic levels.

## MATERIALS AND METHODS

### Cell cultivation and DNA extraction

*L. plantarum* GX17, a probiotic, is isolated from the gut of healthy chicks and maintained by the Key Laboratory of Veterinary Biotechnology at the Guangxi Veterinary Research Institute, Guangxi, China. The glycerol stocks of *L. plantarum* GX17 were activated on a solid Man Rogosa Sharpe (MRS) agar plate and then inoculated into liquid MRS medium for 24 h at 37°C with shaking at 150 rpm. Subsequently, 1 mL of the seed culture was transferred into 100 mL of MRS liquid medium and incubated overnight at 37°C and 150 rpm. The bacterial culture was stored at 4°C. Cells of *L. plantarum* GX17 were collected by centrifugation, and high-quality genomic DNA was extracted and purified using a Qiagen DNA extraction kit.

### Sequencing library construction and sequencing

The genomic library construction, sequencing, and assembly were conducted by Personalbio (Shanghai, China). Sequencing libraries were prepared using 1 µg of genomic DNA, which was fragmented using Covaris. The DNA fragments' sticky ends were converted to blunt ends using an End Repair Mix, and A-tailing was performed on the 3′ ends of all fragments to facilitate the ligation of index adapters. DNA fragments with adapters on both ends were selectively enriched through PCR, simultaneously amplifying the DNA library. The library was quantified using PicoGreen, and samples were pooled in equimolar ratios. Whole-genome shotgun sequencing was conducted on an Illumina MiSeq (32) platform using a paired-end (2 × 250 bp) sequencing approach, with a library insert size of 400 bp.

The sequencing NGS run throughput was 375 G for one lane, with an output of 400 G. Following sequencing, the data were processed, and quality control was performed using FastQC (33) (http://www.bioinformatics.babraham.ac.uk/projects/fastqc/).

The FASTQ files for this result were coded using Illumina version 1.8+, with 97.84% Q20 and 93.7% Q30. AdapterRemoval (https://github.com/MikkelSchubert/adapterremoval) was employed to eliminate adapter sequences at the 3′ ends, and SOAPec (https://help.rc.ufl.edu/doc/SOAPec) was utilized for error correction of all reads based on Kmer frequency (34, 35). The data were assembled using HGAP (v4 http://www.pacb.com/devnet/) and CANU (https://canu.readthedocs.io/en/latest/) to achieve optimal assembly results, which were further refined by local gap filling and base correction with GapCloser software. A genome circular map was generated using CGView (36, 37) (http://stothard.afns.ualberta.ca/cgview_server/).

## Genome sequence analysis

Gene prediction was conducted using GeneMarkS software, while rRNAs and tRNAs within the genome were identified using RNAmmer 1.2 and tRNAscan-SE 2.0.4 software, respectively. The protein sequences of predicted genes were compared against the NCBInr protein database and the COG protein database. Analysis of enzymes related to carbohydrate metabolism was performed using the CAZy database.

## Toxicity factor analysis

Protein coding sequences were compared to amino acid sequences in the virulence factor database using BLAST, with a cutoff E-value of 1e-5, sequence identity over 60%, and sequence length ratio not less than 70%, with gap length less than 10% of the comparison sequence length. In this study, we employed the VF analyzer platform of the Virulence Factors Database (VFDB) (http://www.mgc.ac.cn/VFs/) to annotate and analyze the whole-genome sequences.

## Comparative genomics analysis

### Ortholog clustering analysis

The complete genome sequences of five *L. plantarum* strains used for comparative analysis were obtained from the NCBI database (Table 1). These five strains of *L. plantarum* have good stress resistance in certain specific environments, so they were selected for comparative genomics studies with *L. plantarum* GX17, which has good stress resistance. First, download the protein sequence of the reference genome, filter according to the length of the protein sequence, and remove sequences with a sequence length of less than 50 amino acids. Merge all protein sequences to be analyzed into one file, build a database based on this data set, and use this data set as a query to perform all-VS-all blastp analysis. The threshold for series comparison is set to 1e-10. The sequence alignment results were processed using orthoMCL (version 2.0.8) software (38). The length of the sequence alignment was set to 70%. OrthoMCL was used to cluster the gene family, and clustering 1 was set to 1.5. Finally, a self-made Perl script was used to organize and count the clustering results.

**TABLE 1**  Genomic information involved in the comparison

|  | GenBan number | Genes | Stress resistance |
|---|---|---|---|
| *L. plantarum* KLDS1.0391 | CP019351.1 | 2,691 | It has high resistance to gastrointestinal stress and high adhesion ability to intestinal epithelial cells (Caco-2) (39) |
| *L. plantarum* SPC-SNU 72-2 | CP050805.1 | 2,946 | It is resistant to gastric acid and bile salts and adheres well to colonic epithelial cells (40) |
| *L. plantarum* WCFSI | NC_004567.2 | 3,108 | It is resistant to gastric acid and bile salts, and oxidative stress (41) |
| *L. plantarum* CAUH2 | NZ_CP015126.1 | 2,917 | Oxidative stress resistance (42) |
| *L. plantarum* K25 | CP020099.1 | 2,783 | Cold resistance (43) |

### Collinearity analysis

Download the reference genome sequence. In order to align the start sites of all genomes, first adjust the start sequence of all the other genomes using one genome as a reference. Then Mauve (version 2.3.1) was used to construct and obtain the sequence alignment results of this genome and the reference genome (44).

## Phenotypic assays

### Bacterial solution preparation

The frozen *L. plantarum* GX17 was activated and inoculated into MRS liquid culture medium, cultured at 37℃ overnight, centrifuged at 12,000 rpm/min for 10 min, the supernatant was discarded, and the bacteria were resuspended in the same volume of phosphate-buffered saline (PBS) buffer.

### Acid resistance and bile salt tolerance test

The 10% inoculum was inoculated into a PBS buffer with pH 2, 2.5, 3, and 3.5, and bile salts at concentrations of 0.03%, 0.1%, 0.2%, and 0.3%, respectively. Sampling was performed at 2-h intervals to enumerate the number of colonies on the inoculated plates and to ascertain the change in the number of surviving bacteria from 0 to 6 h. The inoculation was conducted concurrently with the inoculation.

### Simulated gastrointestinal fluid resistance test

Artificial simulated gastrointestinal fluids were prepared according to Lee et al. (45). The bacteria were inoculated into the simulated intestinal fluid or simulated gastric fluid at a 10% inoculation rate. Samples were taken at 0, 30, 60, 90, and 120 min for colony counting to observe the changes in viable bacterial counts. Each group was performed in triplicate.

### Temperature sensitivity test (high temperature/low temperature)

A 10% inoculation volume was then transferred to 5 mL of PBS and subjected to water baths at 37℃, 40℃, 60℃, 70℃, 80℃, and 90℃ for 30 min. Previous experiments have indicated that the optimal growth temperature for *L. plantarum* GX17 is 37℃; hence, the group treated at 37℃ is used as the control group in this experiment. Samples were taken for viable bacterial counts after each treatment, with three replicates per group.

A 1% inoculation volume was then transferred to 5 mL of MRS liquid medium and incubated at 0℃, 5℃, 10℃, and 15℃ for 48 h. Samples were taken for viable bacterial counts after the incubation period, and the survival rate was calculated. Each group was performed in triplicate.

### Growth test of the strain at different salinities

A 1% inoculation volume was then transferred to MRS liquid medium containing 50, 100, and 150 g/L of NaCl and incubated at 37℃ for 24 h. After incubation, viable bacterial counts were performed, and the survival rate was calculated. Each group was performed in triplicate.

### Determination of the antioxidant activity of the strain

After the activation of *L. plantarum*, it was inoculated into a liquid culture medium and incubated overnight. The culture was then centrifuged at 12,000 r/min for 10 min, and the supernatant was collected. The supernatant was filtered through a 0.22 µm filter to remove cellular debris, and the filtrate was stored at 4℃ for further use.

### DPPH radical scavenging activity assay

A volume of 1 mL of the supernatant was pipetted, followed by the addition of 2 mL of a 2,2-Diphenyl-1-picrylhydrazyl (DPPH) anhydrous ethanol solution (0.2 mmol/L). The

mixture was then vortexed and allowed to stand for 30 min at room temperature, in the absence of light. Centrifugation was performed at 8,000 rpm for 10 min, after which the absorbance of the supernatant was determined at 517 nm.

### Superoxide anion radical ($O_2\cdot$) scavenging activity assay

A total of 3.4 mL of Tris-HCl solution (50 mmol/L, pH = 8.2) and 0.5 mL of pyrogallol solution (50 mmol/L) were mixed with 1 mL of the sample. After thorough mixing, the mixture was incubated in a 25℃ incubator for 4 min. The reaction was then terminated by the addition of 0.1 mL of HCl solution (8 mol/L), and the absorbance at 325 nm was measured.

### Hydroxyl radical ($\cdot$OH) scavenging activity assay

One milliliter of o-diazaphene (2.5 mmol/L) was taken, 1 mL of PBS (pH = 7.4) and 0.5 mL of the sample were added, and after mixing, 1 mL of $FeSO_4$ solution (2.5 mmol/L) was added, followed by adding 0.5 mL of $H_2O_2$ (20 mmol/L), and a water bath at 37℃ for 1 h.

### Determination of total reducing power

A solution of 0.5 mL of potassium ferricyanide (1% wt/vol), 0.5 mL of PBS solution (pH = 7.4), and 0.5 mL of the sample should be prepared and mixed thoroughly. This solution should then be placed in a water bath at 50℃ for 20 min, after which 0.5 mL of trichloro-acetic acid (10% wt/vol) should be added, and the solution should be centrifuged at 4,000 rpm for 10 min. The supernatant should be pipetted and mixed with the $FeCl_3$ solution (0.1% wt/vol) in equal volume. The absorbance value at 700 nm should then be tested.

## Quantitative RT-PCR analysis

In order to analyze the expression changes of key genes under stress conditions, one key gene per type was randomly selected and subsequently analyzed via quantitative real-time polymerase chain reaction (qRT-PCR). For each group, total RNA was extracted from *L. plantarum* GX17 using Trizol (CWBIO) according to the manufacturer's instructions, and 1 mg was used as a template for first-strand cDNA synthesis using the HiFiScript cDNA Amplification System (CWBIO). *16S ribosomal RNA* was incorporated as an endogenous control. The specific primers utilized in the qRT-PCR assays are enumerated in the accompanying Table 2. All reactions were subjected to qRT-PCR in triplicate using the SYBR Green Detection System and Light Cycler 96 Real-Time PCR System. Normalization of the cycling threshold (CT) values for each sample against the reference gene primers was conducted, and the calculation of relative changes in gene expression was performed using the 2-ΔΔCT method.

## Statistical analysis

Three replicates were conducted for each experiment. The data were then collated and subjected to statistical analysis using Excel and SPSS 26 statistical software. One-way (analysis of variance) multiple comparison analyses were performed, and GraphPad Prism 9.5.0 software was employed for plotting.

## RESULTS

### Basic features of *L. plantarum* GX17 genome

As depicted in Fig. 1, the *L. plantarum* GX17 genome comprises a single circular chromosome and four circular plasmids. The genome's fundamental characteristics are detailed in Table 3. The assembly effect of the complete sequence was evaluated. The chromosome genome spans 2,952,198 base pairs (bp) with an average GC content of 44.53%. The four circular plasmids measure 12,458 bp (plasmid 1) with 39.41% GC

**TABLE 2** Specific primers used in qRT-PCR assays

| Gene name | Location | | Primer sequence | Amplification length (bp) |
|---|---|---|---|---|
| 16S ribosomal RNA | chr_7 | F | GCTCGTGTCGTGAGATGTT | 150 |
| | | R | TGTAGCCCAGGTCATAAGG | |
| cspL | chr_28 | F | UGGUACAGUAAAAUGGUUCAA | 152 |
| | | R | CCUGUUCUUCAUCAUAAGU | |
| hsp18 | chr_2088 | F | GATCTACTAAAGCCCACCAAA | 191 |
| | | R | GCCCGAATAGTTAGCCAT | |
| asp23 | chr_694 | F | GTCTAGCTTCACGCAATGTT | 175 |
| | | R | CGCATGTCCTTACCATATTCA | |
| nhaC | chr_163 | F | CTAACTAAGCGATTGAAAGGT | 171 |
| | | R | GACTCGACTGAGGGCTAAG | |
| bsh | chr_55 | F | GGCCAAGCAACCTATACTGA | 146 |
| | | R | TATTCTAACGGAACGGTCTGT | |
| SH1215 | chr_2149 | F | GGGAAGTCCGAAACCAATTAT | 132 |
| | | R | CGCTGCACATACGTTGTAACC | |

content, 11,921 bp (plasmid 2) with 37.66% GC content, 2,968 bp (plasmid 3) with 38.24% GC content, and 2,411 bp (plasmid 4) with 38.66% GC content, respectively. The entire genome encompasses 1,771 protein-coding genes, 21 rRNA genes, 77 tRNA genes, and 122 pseudogenes.

## Genomic functional annotation and analysis of *L. plantarum* GX17

The *L. plantarum* GX17 genome's predicted protein sequences were annotated using databases such as Kyoto Encyclopedia of Genes and Genomes (KEGG), Clusters of Orthologous Groups of proteins (COG), Evolutionary Genealogy of Genes: Non-supervised Orthologous Groups (eggNOG), Gene Ontology (GO), and CAZy. When multiple annotation results were available, the annotation with the best evidence was selected.

## KEGG and COG database annotations of *L. plantarum* GX17

A total of 1,436 genes were annotated in the KEGG database (Fig. 2), categorized into six major classes encompassing 38 subclasses. The annotations included 75 genes related to cellular processes, 200 to environmental information processing, 194 to genetic information processing, 81 to human diseases, 845 to metabolism, and 32 to organismal systems, with the largest proportion, 792 genes, involved in metabolism. This included 237 genes related to carbohydrate metabolism and 163 to amino acid metabolism. The environmental information processing category notably featured 135 genes associated with membrane transport.

Applying the COG database for functional annotation revealed that 2,414 of the 2,783 coding sequences (CDSs) could be classified into 18 COG categories (Fig. 3). Around 20.98% of the genes were of unknown function and labeled as putative proteins. Known functions predominantly included transcription (240 genes), carbohydrate transport and metabolism (204 genes), and amino acid transport and metabolism (192 genes). Approximately 13.26% of the genes could not be matched with any COG database entries.

## CAZy database annotation of *L. plantarum* GX17

CAZy database annotation identified 102 genes in *L. plantarum* GX17 (Fig. 4), with 39 being glycoside hydrolases (GHs), and 29 glycosyl transferases (GTs), including 16 carbohydrate esterases (CEs). GHs, constituting 46.43% of the carbohydrate-related genes, predominantly act on 1,4-α-D-glucosidic bonds in polysaccharides, releasing

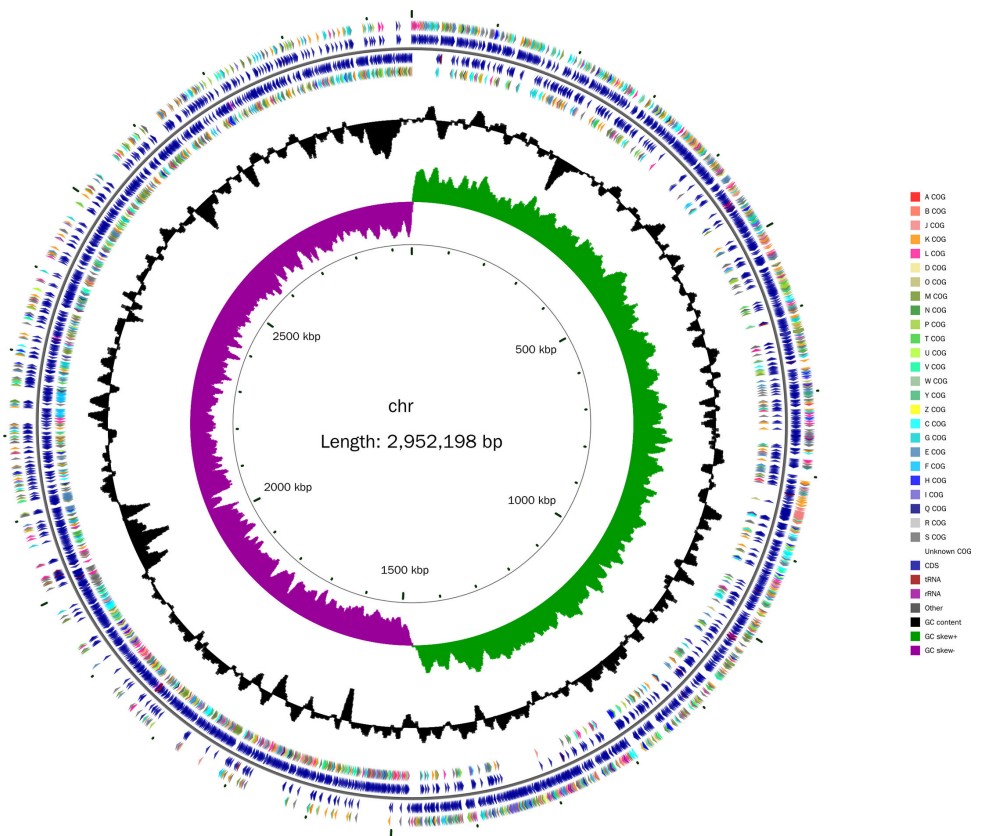

**FIG 1** Circular graph of *L. plantarum* GX17 complete genome. From inside to outside, the first circle represents the scale; the second circle represents GCSkew; the third circle represents the GC content; the fourth and seventh circles represent the COG to which each coding sequence (CDS) belongs; the fifth and sixth circles represent the positions of CDS, tRNA, and rRNA on the genome.

energy for bacterial activities. GT-annotated genes, making up 34.52% of the carbohydrate metabolic genes, ranked second in annotation quantity.

## Toxicity factor analysis

Protein coding sequences were compared to amino acid sequences in the virulence factor database using BLAST, with a cutoff E-value of 1e-5, sequence identity over 60%, and sequence length ratio not less than 70%, with gap length less than 10% of the comparison sequence length. The screening results are summarized in Table 4.

## Two-component systems

Four pairs of genes (chr_1487 and chr_1488, chr_1956 and chr_1957, chr_2473 and chr_2474, chr_2745 and chr_2746) belong to the two-component regulatory system (TCS), essential for sensing and responding to environmental changes.

**TABLE 3** Fundamental characteristics of *L. plantarum* GX17

| Sample | Seq ID | Seq length (bp) | GC content (%) | Seq type |
|---|---|---|---|---|
| *Lactiplantibacillus plantarum* GX17 | chr | 2,952,198 | 44.93 | Circular |
| | plasmid 1 | 12,458 | 39.41 | Circular |
| | plasmid 2 | 11,921 | 37.66 | Circular |
| | plasmid 3 | 2,968 | 38.24 | Circular |
| | plasmid 4 | 2,411 | 38.66 | Circular |

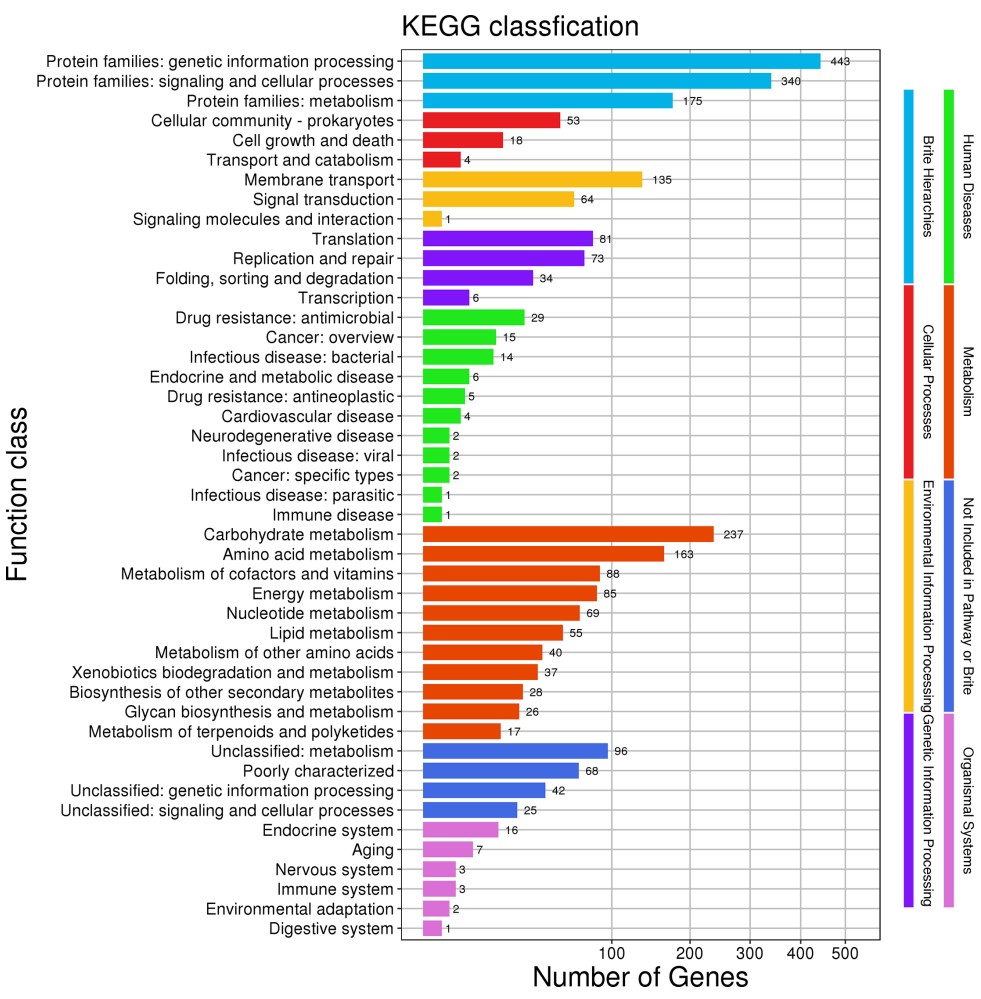

**FIG 2** *L. plantarum* GX17 KEGG database annotation results.

## Antistress gene analysis of *L. plantarum* GX17

By analyzing the whole-genome sequence of *L. plantarum* and comparing it with other published species known for strong stress resistance, potential stress resistance genes were identified. Analysis revealed 50 antistress genes across seven categories: temperature, phage, acid, $Na^+/H^+$, bile, adhesion, and antioxidant activity (Table 5). The universal stress protein (UspA), with 10 encoding genes, had the highest count, followed by eight genes encoding the $Na^+/H^+$ antiporter.

## Comparative genomic analysis on *L. plantarum* GX17

The results of the gene family analysis demonstrated that the six strains of *L. plantarum* shared 2,120 genes (Fig. 5), with 96 genes being unique to *L. plantarum* GX17. The differences in the number of unique genes among the six strains were minimal, which may be attributed to the fact that these samples exhibited robust resistance to stress. With the exception of the untagged genes, the unique genes of *L. plantarum* GX17 are related to carbohydrate transport and metabolism. Of these, the gene chr_2428, which is involved in inorganic ion transport and metabolism, is a $Na^+/H^+$ antiporter-related gene; this gene is related to maintaining intracellular ion homeostasis and improving salt and drought tolerance. This suggests that the strain may be resistant to salt environments (Table 6).

A comparison of the chromosomal genomes of six strains of *L. plantarum* using Mauve revealed that the degree of conservation varied among the strains and that

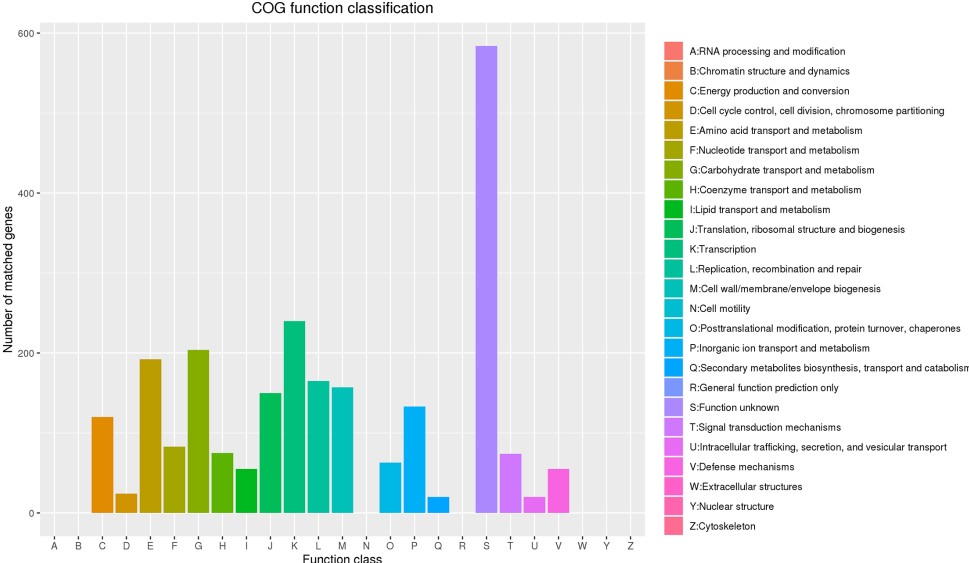

**FIG 3** The number of matched genes assigned in cluster orthologous groups (COGs) in *L. plantarum* GX17.

there were genomic structural differences (Fig. 6). The densest connecting line between KLDS1.0391 and SPC-SNU72-2 indicated that the highest homology was between the strains KLDS1.0391 and SPC-SNU72-2, which originated from the same isolate and are positioned closest on the phylogenetic tree. The denser connecting line between *L. plantarum* GX17 and CAUH2 indicates that the homology between these two strains is higher than that observed between other *L. plantarum* strains. Furthermore, additional partial inversions were observed in the distribution of homologous genes between *L. plantarum* GX17 and KLDS1.0391. However, in the case of the remaining four *L. plantarum* strains, only individual homologous blocks underwent inversion, suggesting evolutionary differences between *L. plantarum* GX17 and KLDS1.0391. The genomic differences with the reference strains indicate that the genomes were subjected to recombination and transfer during the evolutionary process.

## Phenotypic results

### *L. plantarum* GX17 acid tolerance test

The number of surviving bacteria of *L. plantarum* GX17 exhibited a decline over time when maintained in PBS at pH 2.0–3.5. A reduction in pH was also observed to result in a decline in the number of surviving bacteria over the same period of time. The logarithmic value of *L. plantarum* decreased to 0 after 4 h of treatment with a PBS solution at pH 2.0, but remained well tolerated after 6 h of exposure to a PBS solution at pH 2.5–3.5 (Fig. 7a).

### *L. plantarum* GX17 bile salt tolerance test

The number of viable bacteria declined over time when *L. plantarum* GX17 was treated with bile salts at concentrations ranging from 0.03% to 0.3%. Furthermore, the decrease in the number of viable bacteria was found to be statistically significant with the increase in bile salt concentration. The logarithmic value of the number of live bacteria of the strain decreased significantly from 11.56 to 4.87 in the first 2 h when treated with 0.3% porcine bile salts. There was a further decrease from 4.87 to 2.93 between 2 and 6 h, although this represented a relatively minor decline in comparison to the initial 2-h period. This suggests that *L. plantarum* GX17 is well tolerated by bile salts and exhibits robust probiotic properties (Fig. 7b). These attributes enable the strain to traverse the

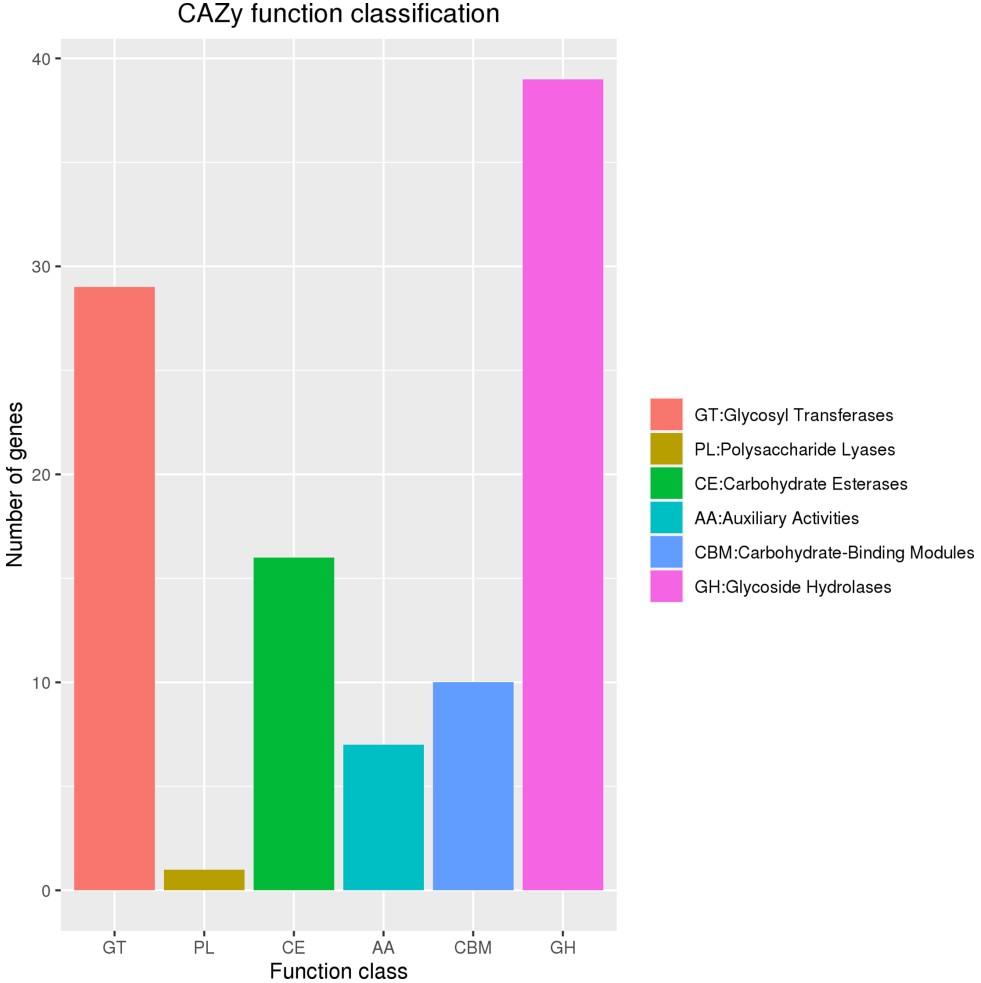

**FIG 4**  *L. plantarum* GX17 CAZy database.

gastrointestinal tract and colonize the intestinal lumen, where it can exert its beneficial effects.

### Tolerance of L. plantarum GX17 in simulated artificial gastrointestinal fluids

Following a 2-h exposure to simulated artificial gastric and intestinal fluids, the logarithmic value of the number of viable bacteria of the *L. plantarum* GX17 strain exhibited a decline compared to the initial ratio and logarithmic value. The decline of 7.22 and 1.49, respectively, indicates that the *L. plantarum* GX17 strain was well tolerated by gastric and intestinal fluids (Fig. 7c).

### Effect of temperature on the growth of *L. plantarum* GX17

Following the high-temperature treatment of *L. plantarum* GX17 for 30 min, the logarithmic value of the number of viable bacteria of the strain exhibited a decline in comparison to that of the 37°C treatment group. However, three logarithmic values of the number of viable bacteria were observed after 30 min of treatment at 70°C, and the survival of bacteria was also evident following treatment at 80°C. This suggests that *L. plantarum* GX17 exhibits robust thermotolerance (Fig. 7e).

The survival rate of *L. plantarum* GX17 was 29%, 36%, 47%, and 15,600% after 48 h of treatment at 0°C, 5°C, 10°C, and 15°C, respectively. This indicates that the strain is also

**TABLE 4** Statistics of GX17 virulence factors of *L. plantarum*

| VFDB ID | ORF name | VFDB name | Gene symbol | Function |
|---|---|---|---|---|
| VFG012095(gb\|WP_003435012) | chr_514 | VF0594 | *groL* | Involved in adhesion or invasion of various target cells or tissues |
| VFG000964(gb\|WP_010922799) | chr_539 | VF0244 | *hasC* | Plays an adhesion and protection role and can also act as a molecular mimic to evade the host immune system during infection (46) |
| VFG000077(gb\|NP_465991) | chr_562 | VF0074 | *clpP* | Serine protease involved in proteolysis and is required for growth under stress conditions (47) |
| VFG000080(gb\|NP_464522) | chr_925 | VF0073 | *clpE* | An ATPase required for prolonged survival at 42 degrees. Acts synergistically with ClpC in cell division (48) |
| VFG048830(gb\|WP_014907233) | chr_1142 | VF0560 | *Gnd* | Polymorphic gene encoding 6-phosphogluconate dehydrogenase (49) |
| VFG002190(gb\|WP_002362225) | chr_1564 | VF0361 | *uppS/cpsA* | Immune modulation; Antiphagocytosis contributes to host immune evasion (50) |
| VFG046465(gb\|WP_003028672) | chr_1623 | VF0460 | *tuf* | Produces extracellular enzymes and adheres to host cells (51) |

well tolerated at low temperatures and that the strain can grow at 15℃ with a good growth status (Fig. 7f).

### Effect of different osmotic pressures on the growth of L. plantarum GX17

In comparison with the control group (salt concentration = 0 g/L), the number of surviving bacteria of *L. plantarum* GX17 demonstrated a distinct decline with the increase of salt concentration in the medium. At salt concentrations of 50, 100, and 150 g/L, the number of surviving bacteria exhibited a reduction of 1.2, 5.9, and 6.3 logarithmic values, respectively. *L. plantarum* GX17 demonstrated the capacity to survive for 24 h in a high-salt medium with a salt concentration of 150 g/L, indicating its ability to tolerate

**TABLE 5** The antistress proteins of *L. plantarum* GX17 genome

| Stresses | Product | Locus | Function |
|---|---|---|---|
| Temperature | Cold shock protein | chr_28, chr_746, chr_877, | Regulates cold shock response and responses to various exogenous stress conditions (hyperosmotic pressure, starvation, antibiotics, organic solvents, etc.) |
| | Heat shock protein | chr_110, chr_2088, chr_2596 | Help cells resist various adverse factors, such as high temperature, hypoxia, oxidative stress, etc. (52) |
| Phage | Phage shock protein C PspC | chr_107, chr_534 | Helps to ensure the integrity of the cell membrane under environmental stress and maintain the energy status of the cell under stress conditions (53) |
| Acid | Alkaline shock protein | chr_694, chr_695, chr_1189 | Proteins that accumulate in the soluble protein fraction after alkaline shock (54) |
| $Na^+/H^+$ | $Na^+/H^+$ antiporter | chr_163, chr_589, chr_663, chr_2045, chr_2092, chr_2428, chr_2429, chr_2584 | Maintain intracellular ion homeostasis (55) and improve salt (56) and drought tolerance |
| Bile | Choloylglycine hydrolase | chr_55, chr_57, chr_2024, chr_2604 | Hydrolyzes the amide bond between glycine or taurine and the steroid nucleus of bile acids, countering the effects of bile acids (57) |
| Adhesion | Fibronectin-binding protein | chr_44, chr_178, chr_1355 | Binds specifically to fibronectin and participates in the adhesion of bacteria to the ECM of host cells (58) |
| Antioxidant activity | Thioredoxin reductase | chr_542 | Resist oxidative stress and regulate redox balance (59) |
| | Thioredoxin | chr_203, chr_542, chr_1753, chr_2061, chr_2653 | |
| | NADH oxidase | chr_122, chr_1056, chr_547 | Major antioxidant defense enzyme to resist oxidative stress (60) |
| | Oxidoreductase | chr_587, chr_857, chr_878, chr_1338, chr_1456 | Catalyzes the redox reaction between two molecules |
| | Universal stress protein UspA | chr_2071, chr_2149, chr_2264, chr_2358, chr_2765, chr_880, chr_972, chr_1281, chr_1317, chr_1809 | Can participate in the resistance to a variety of abiotic and biotic stresses (61, 62) |

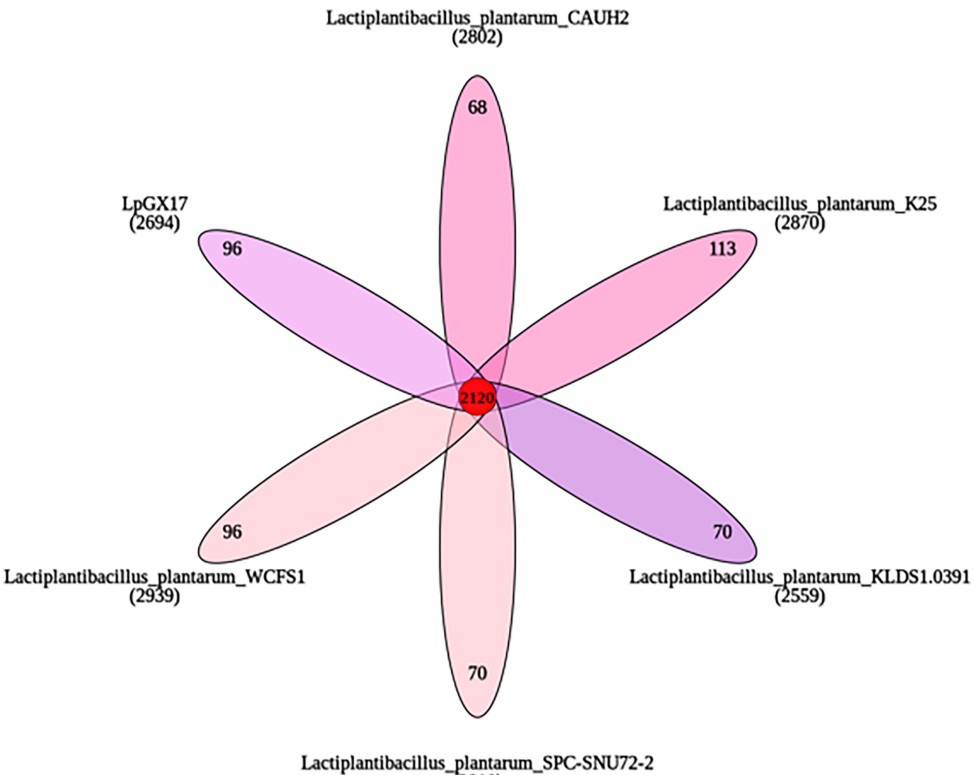

**FIG 5** Gene family analysis.

fluctuations in osmotic pressure, and to withstand unfavorable environmental influences (Fig. 7d).

### *In vitro* antioxidant activity of *L. plantarum* GX17

*In vitro* studies have demonstrated that *L. plantarum* GX17 is capable of scavenging 13% of superoxide anion. Furthermore, the bacterium exhibits a robust scavenging ability for DPPH radicals and hydroxyl radicals, with scavenging rates of 94.39% and 62.31%, respectively. The total reducing power of *L. plantarum* GX17 was determined using the potassium ferricyanide reduction method, and the results demonstrated that the total reducing power of *L. plantarum* GX17 reached 95.63% *in vitro* (Fig. 7g). These findings indicate that *L. plantarum* GX17 has a robust reducing capacity *in vitro*, particularly in scavenging hydroxyl radicals and DPPH radicals. It has the potential to mitigate cellular oxidative damage and the associated pathological processes, including cellular senescence.

### Expression analysis of key genes

As shown in Fig. 8a, the expression level of the *cspL* gene in *L. plantarum* GX17 was significantly upregulated to 1.6-fold of the control group after 48 h of cold stress at 0°C. Under thermal stress conditions, the expression of the *hsp* gene exhibited differential upregulation, with significant increases to 1.2- and 1.3-fold of control levels following treatment at 60°C and 70°C, respectively (Fig. 8b). Acidic stress (pH 3.0 and 3.5) induced 1.2- and 1.3-fold upregulation of *asp23* gene expression (Fig. 8c). The transcriptional response of *nhaC* to osmotic stress demonstrated a concentration-dependent biphasic pattern, peaking at 1.6-fold induction under low salinity conditions (Fig. 8d). Notably, bile acid exposure (0.1%–0.3%) triggered progressive upregulation of *bsh* (encoding bile salt hydrolase) to 1.2-, 1.5-, and 1.8-fold, respectively (Fig. 8e). Both *asp23* and *bsh*

**TABLE 6** Function and number of COGs in *L. plantarum* GX17-specific genes

| COG categories | Categories functions | Locus | Number |
|---|---|---|---|
| C | Energy production and conversion | chr_41, chr_250, chr_803 | 3 |
| E | Amino acid transport and metabolism | chr_622, chr_1996 | 2 |
| G | Carbohydrate transport and metabolism | chr_230, chr_231, chr_1304, chr_1628, chr_2224, chr_2373, chr_2735 | 7 |
| K | Transcription | chr_2724 | 1 |
| L | Replication, recombination, and repair | chr_1880, chr_1915 | 2 |
| M | Cell wall/membrane/envelope biogenesis | chr_2191 | 1 |
| P | Inorganic ion transport and metabolism | chr_2428 | 1 |
| S | Function unknown | chr_704, chr_960, chr_1248, chr_1872, chr_1873, chr_1876, chr_1911, chr_2622 | 8 |
| T | Signal transduction mechanisms | chr_241, chr_1054 | 2 |

exhibited upregulated expression following simulated gastrointestinal fluid challenge (Fig. 8f). Furthermore, the UspA gene (*SH1215*) displayed consistent upregulation across all tested stress conditions, suggesting its pivotal role in general stress response.

## DISCUSSION

With the continuous deepening of research on LAB, they are widely used in food fermentation, industrial biotechnology, and play a promising role in medicine as probiotics, immunomodulators, and drug delivery systems. Therefore, the impact of pressure on LAB has become the subject of much research. In fact, any conditions that deviate from optimal environmental conditions, such as temperature, osmotic pressure and pH shocks, ultraviolet radiation, various oxidants, etc., are considered pressure conditions (63). Therefore, this study aims to investigate the stress resistance of *L. plantarum* GX17, conduct in-depth research on its gene library, explore its stress resistance mechanism, evaluate the probiotic properties of this probiotic by combining genotype and phenotypic analysis, and explain the rationality of this property at the molecular level.

In non-pathogenic bacteria, the presence of prophage sequences significantly contributes to their adaptation to specific environments. The genome of *L. plantarum* exhibits considerable plasticity, primarily due to horizontal gene transfer facilitated by mobile genetic elements such as phages, plasmids, and transposons (64). Prophages are nucleic acids integrated into the host bacterial chromosome following the invasion

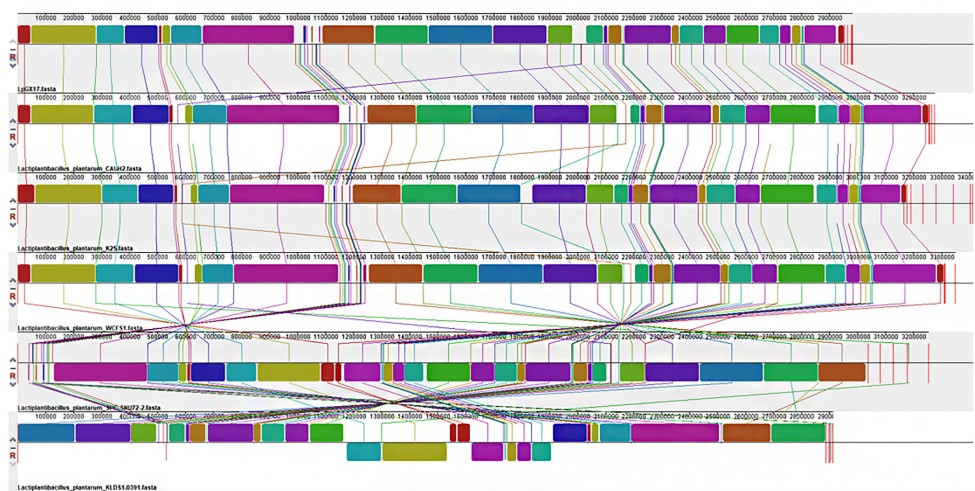

**FIG 6** Mauve whole gene sequence comparison.

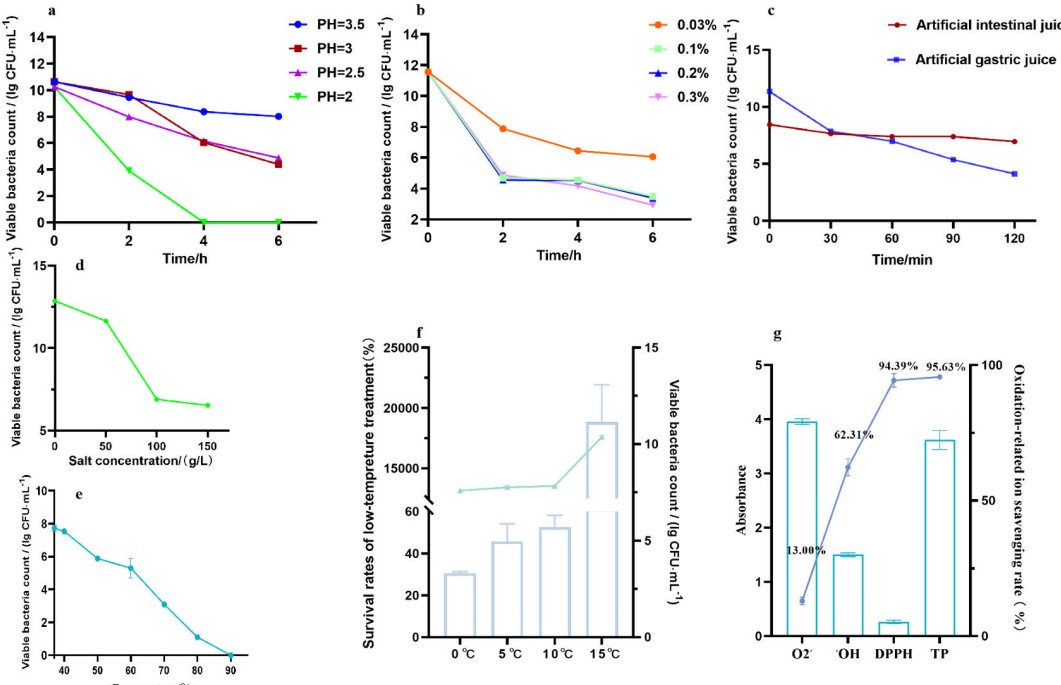

**FIG 7** The results of the *in vitro* biological activity assay of *L. plantarum* GX17 are presented herewith. (a) Effects of different pHs on the growth of *L. plantarum* GX17 strain. (b) Effects of different concentrations of bile salt on the growth of *L. plantarum* GX17 strain. (c) Effects of artificial gastrointestinal fluid on the growth of *L. plantarum* GX17. (d) Effects of different osmotic pressures on the growth of *L. plantarum* GX17 strain. (e) Effects of high temperature on the growth of *L. plantarum* GX17 strain. (f) Effects of low temperature on the growth of *L. plantarum* GX17 strain. (g) Determination of antioxidant activity of *L. plantarum* GX17 strain.

by certain phages, which can be either virulent or mild. Virulent phages, like the *E. coli* T4 phage, engage in a lytic cycle that involves the attachment to a specific bacterial receptor, DNA injection, replication, assembly of new virus particles, and host lysis to release the progeny viruses. Conversely, mild phages, such as the *E. coli* λ phage, may initiate infection similarly but can enter a lysogenic cycle where viral gene expression is suppressed by phage-encoded repressors, leading to the integration of dormant prophages into the host chromosome or the formation of self-replicating plasmids. Lysogenic cells become immune to further infections by the same phage due to lysogenic repressors (65).

The KEGG annotation suggests that GX17 has robust carbohydrate and amino acid metabolism capabilities and efficient membrane transport systems. COG annotations highlight a significant number of genes involved in carbohydrate transport and metabolism, particularly in sugar biosynthesis from nucleotides, indicating the strain's potent sugar biosynthesis and transport capabilities. The CAZy database, dedicated to enzymes that synthesize or degrade complex carbohydrates and glycoconjugates, reveals that in *L. plantarum* GX17, GHs are the most annotated, followed by glycosyl transferases, which form glycosidic bonds and transfer sugars to specific acceptors, thus participating in various physiological processes (66). Glycoesterases, ranking third in annotation, catalyze the de-esterification of carbohydrate substrates. Although *L. plantarum* GX17 has fewer annotated auxiliary enzyme genes, those present are crucial for the redox activity of carbohydrates.

Virulence factors (VFs), encoded by genes on chromosomes or mobile genetic elements, include toxins, attachment proteins, protective surface molecules, and pathogenic hydrolases (67). Seven virulence-related genes were identified in *L. plantarum* GX17. Most of them are adhesion or non-classical virulence factors. Adherence to human tissues and the gastrointestinal tract is a pivotal virulence attribute for

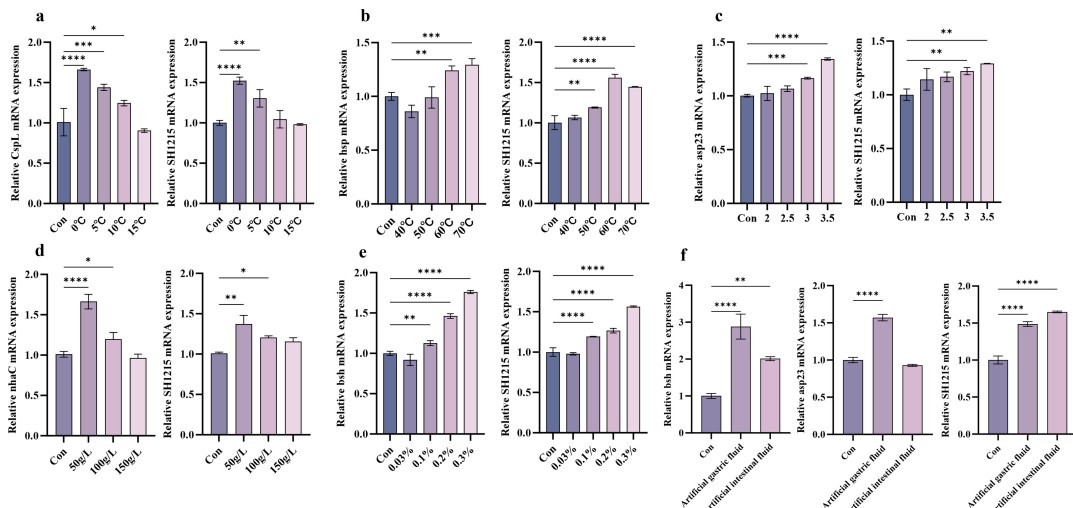

**FIG 8** Expression of key genes in *L. plantarum* GX17 under different treatment conditions. (a) Gene expression of *L. plantarum* GX17 under low-temperature treatment conditions. (b) Gene expression of *L. plantarum* GX17 under high-temperature treatment conditions. (c) Gene expression of *L. plantarum* GX17 under different pH treatment conditions. (d) Gene expression of *L. plantarum* GX17 under different osmotic pressure treatment conditions. (e) Gene expression of *L. plantarum* GX17 under different concentrations of bile salt treatment conditions. (f) Gene expression of *L. plantarum* GX17 after artificial gastrointestinal fluid treatment.

pathogenic microorganisms and a positive characteristic for probiotics, which is integral to the criteria for assessing novel probiotic strains (68). Adherence to intestinal epithelial cells and competitive exclusion of pathogens enhance probiotic persistence and intestinal colonization (69). Streptococcus thermophilus TK-P3A VFs enhance probiotic survival in the GI tract (70–77), while *L. plantarum* GX17 VFs are linked to adaptation and attachment in harsh conditions, potentially increasing bacterial resistance. Characteristic VFs like the fsr locus in *Enterococcus faecalis* TK-P4B and *Enterococcus faecium* TK-P5D may pose consumption risks (78–81). However, the *L. plantarum* GX17 demonstrated safety, improving growth performance in chicks after 42 days of feeding (24). To date, there is no Lactobacillus-specific gene database relevant to safety assessments, and existing databases tend to focus primarily on pathogens. Misuse of the VFDB database may lead to misleading results in the safety assessment of lactobacilli (82). Although 126 so-called virulence genes were found in *L. plantarum* JDM1, these genes do not actually pose a safety problem because they do not encode toxins or invasion proteins (83). This shows that more precise and Lactobacillus-specific tools and databases are needed when evaluating the safety of lactobacilli.

The surge in probiotic research has fueled the production of functional foods and drugs enriched with these beneficial microorganisms. Originating from the intestinal tract, probiotics encounter significant challenges in maintaining viability throughout processing, storage, and their journey through the gastrointestinal tract to their site of action within the human body. These bacteria face various stresses, including temperature fluctuations, acid exposure, bile salts, osmotic conditions, and oxidative stress during product preparation and gastrointestinal transit. Nonetheless, like all bacteria, probiotics possess a broad spectrum of molecular mechanisms to counteract the environmental stresses frequently encountered both during processing and post-ingestion.

TCSs are signal transduction mechanisms often comprising a membrane-bound sensor kinase and a cytoplasmic response regulator activated through histidine-to-aspartate phosphorelay reactions. TCS enables bacteria to respond to environmental stimuli such as redox potential, pH, specific metabolites, stress, light, and antimicrobial peptides (84). In *L. plantarum* GX17, the TCS may mediate responses to a variety of signals and stressors, including acid tolerance, osmotic stress, and bacteriocin biosynthesis, thereby

regulating numerous physiological functions. Furthermore, to adapt to environmental variations, numerous stress-resistant genes have emerged within the *L. plantarum* genome, along with mechanisms for stress adaptation or mitigation.

Cryopreservation is crucial for preserving cell viability, but it can also induce cold stress that affects membrane fluidity, enzyme function, and RNA stability, thereby impacting bacterial survival (85). *L. plantarum* GX17 demonstrated significant cold tolerance *in vitro*, with 36% of the bacteria surviving 48 h at 5℃, a trait comparable to other plant lactobacilli. *L. plantarum,* notably strain L67, also showed strong resistance to cold stress, with 78% survival after a 6 h exposure at 5℃ followed by freeze-thaw conditions (86). Genomic analysis of *L. plantarum* GX17 indicates that the presence of Csp proteins may enhance freezing resistance and survival, which is corroborated by its *in vitro* low-temperature tolerance (87). Csp proteins have been implicated in cold adaptation and survival across various probiotic studies, playing a pivotal role in cellular responses to low temperature stress through multiple mechanisms (87–90). The ability to withstand low temperatures is vital for probiotic stability.

Heat stress impairs key microbial functions, primarily by damaging bacterial membranes, fatty acids, proteins, and ribosomes, as well as causing RNA damage (91, 92). Heat shock proteins (Hsp) are conserved proteins induced by stress, which increase thermal tolerance and adaptability to temperature changes (93). The presence of *hsp* genes in *L. plantarum* GX17 suggests potential for enhanced probiotic survival during high-temperature processing. *In vitro* experiments confirm *L. plantarum* GX17's robust high-temperature tolerance, contrasting with the reduced survival of *L. plantarum* LB5 at 50℃ and 60℃ (94). Notably, *L. plantarum* GX17 maintained 3 log viable bacterial counts after a 30 min exposure to 70℃, outperforming *L. plantarum* K8, which showed high tolerance to a brief 70℃ treatment (95). The thermostable nature of *L. plantarum* GX17 bolsters its stability throughout production, storage, and transportation, particularly in high-temperature conditions, ensuring product efficacy for consumers (96). This trait also improves fermented food resilience to temperature stress, minimizes contamination, promotes faster growth, and enhances lactic acid production during fermentation and drying processes (97).

A significant challenge for probiotics is surviving gastric acid. Both *L. plantarum* DKL3 and JGR2 showed <1 log10 CFU reduction after 3-h exposure to a pH of 3.9 (98). After *L. plantarum* 9010 was exposed to simulated gastric juice *in vitro* for 120 min, the number of viable bacteria decreased by approximately 44% (99). After simulated gastric digestion (2 h) and simulated intestinal digestion (2 h), the number of viable bacteria in *L. plantarum* 9010 decreased by about 8% compared with the end of simulated gastric digestion. Similar results were also found in this study. PBS solution at pH 2.5–3.5 was still well tolerated for 6 h, and a large number of bacteria were still alive after 60 min of treatment in the artificial gastrointestinal tract. These results are consistent that lactobacilli remain active between pH 2.5 and 4.0, and the conclusion is consistent that lactobacilli with strong tolerance have good adaptability to the gastrointestinal environment (94, 100–102). All LAB isolates isolated by RineChristopher Reuben et al. could tolerate increased NaCl concentrations to 6.5%, and all isolates grew very weak at 10.0% NaCl. After incubating *L. plantarum* GX17 in a salt solution with a concentration of 150 g/L for 24 h, $10^3$ bacteria were still alive and were highly salt-tolerant. This phenotypic characterization supports the presence of three genes encoding alkaline shock proteins and eight genes encoding the sodium-proton antiporter ($Na^+/H^+$), crucial for maintaining cellular pH and $Na^+$ homeostasis across various life forms (103, 104). Similar acid tolerance mechanisms have been observed in *L. plantarum* Y44 (105) and *L. amylolyticus* L6 (106) through whole-genome sequence analysis.

Bile, synthesized by liver parenchymal cells, aids in fat digestion and the absorption of fat-soluble vitamins, while its bile salts have antimicrobial properties that can disrupt cellular membranes and cause oxidative DNA damage (107). Genes in *L. plantarum* GX17 encoding choloylglycine hydrolase contribute to its resilience against high bile salt concentrations. Studies have shown that at a bile salt concentration of 2%, the relative

*bsh* gene expression levels of *L. plantarum* 9 and *L. plantarum* 91 were the highest (108). Therefore, consistent with the results of this study, the expression of the *bsh* gene can be considered as a prospective biomarker for screening novel probiotic strains with optimal function in the intestine. Phenotypic assays showed that *L. plantarum* GX17 could survive a 0.3% bile salt concentration for 6 h, maintaining $10^3$ CFU/mL viable bacteria, suggesting the encoded choloylglycine hydrolase aids in coping with high bile salt environments.

The *L. plantarum* GX17 genome contains antioxidant-related genes, including thioredoxin reductase, NADH oxidase, oxidoreductase, and the universal stress protein UspA, which may contribute to its antioxidant capabilities. Similarly, *Bifidobacterium longum* LTBL16 has at least five protein-coding genes linked to antioxidant activity, implying a correlation between gene expression and antioxidant efficiency (109). *L. plantarum* GX17 has shown strong *in vitro* antioxidant properties, with 94.39% DPPH radical scavenging, 62.31% OH radical scavenging, and 95.64% total reducing power, along with a 13% superoxide anion radical rate. In comparison, other *L. plantarum* strains exhibit only 40% DPPH scavenging activity, and *L. plantarum* LB5 at $10^7$ CFU/mL has approximately 10% ABTS free radical scavenging activity. These findings highlight *L. plantarum* GX17's superior antioxidant potential (94). This may be related to the fact that *L. plantarum* GX17 has multiple genes that express antioxidation-related enzymes. This is consistent with other studies, where the entire genome contains antioxidation-related genes and all show antioxidant activity in *in vitro* experiments (110). Studies have shown that *L. plantarum* can effectively reduce protein oxidation in fermented sausages (111) and increase the total phenolic content and antioxidant capacity of fermented pomegranate juice (112). In addition, *L. plantarum* AS1 significantly enhanced the antioxidant capacity of high-fat diet-fed rats by improving lipid peroxidation and antioxidant activity in the colon and plasma (113, 114). These findings suggest that the antioxidant capacity of *L. plantarum* GX17 may become a potential tool for the prevention and treatment of oxidative stress-related diseases.

Interestingly, the genome of *L. plantarum* GX17 encodes genes for fibronectin-binding proteins (found in chr_44, chr_178, and chr_1355) that enhance adhesion to host intestinal epithelial cells and facilitate protein synthesis. It has been demonstrated that FbpA, produced by the probiotic *Weissella cibaria*, reduces *S. aureus* colonization and infection in the mammary glands by inhibiting the formation of *S. aureus* biofilms (109). This suggests that the presence of Fbp proteins not only aids in the probiotic bacteria's colonization but may also play a role in preventing colonization and infection by pathogenic bacteria.

The PspC gene (found in chr_107, chr_534) is present in the genome of *L. plantarum* GX17. This gene plays a key role in the periplasmic stress response (Psp response). The Psp response is a broadly conserved bacterial protective mechanism that defends against both internal and external factors potentially damaging to the cell membrane by monitoring its state and modulating transcriptional responses (115–117). It is posited that the presence of the PspC gene suggests that *L. plantarum* GX17 may use this mechanism to adapt and respond to environmental stress.

We also identified genes encoding the UspA in the genome of *L. plantarum* GX17, a finding seldom reported in other *L. plantarum* strains. In *E. coli*, UspA is a key stress protein, and its overexpression under stress conditions is essential for cell survival (118, 119). UspA deficiency leads to premature senescence of growth-arrested cells, whereas its overexpression promotes growth arrest and upregulates multiple functional proteins (120). Therefore, UspA significantly regulates cellular protein expression and plays a key role in resisting superoxide damage, protecting cells from harmful effects (119). We propose that the expression of the UspA gene in *L. plantarum* GX17 could boost the strain's adaptability in stressful environments. In summary, the discovery of these anti-stress genes in *L. plantarum* GX17 provides compelling evidence of the strain's ability to navigate various stressful conditions.

To elucidate the relationship between gene expression and phenotypic traits under stress, we analyzed the transcriptional responses of key stress-related genes in *L. plantarum* GX17. After 48 h of cold stress, *cspL* expression was significantly upregulated, consistent with its role in cold adaptation (86). Under heat stress (60°C–70°C), *hsp* genes were markedly induced, aligning with previous findings that overexpression of *hsp18.5*, *hsp18.55*, and *hsp19.3* enhances thermotolerance in *L. plantarum* WCFS1 (121), highlighting the importance of heat shock proteins in thermal adaptation. Similarly, under moderate acid stress (pH 3.5), *asp23* expression increased significantly, as reported in *S. aureus* (122). However, under more severe acid stress (pH 2–3), *asp23* expression remained unchanged, possibly due to suppression of the $\sigma^B$-dependent stress response or a shift toward survival mechanisms (123). A comparable trend was observed for *nhaC* under salt stress: its expression increased under low salt but remained stable or decreased under high salt, likely reflecting cellular damage and a reallocation of resources toward survival (124). In contrast, *bsh*, encoding choloylglycine hydrolase, was significantly upregulated under bile salt stress, consistent with its role in enhancing bile tolerance in *Bifidobacterium longum*, *L. plantarum*, and *Limosilactobacillus reuteri* (125). However, this response appears strain-specific, as *Lactobacillus salivarius* did not exhibit *bsh1* induction under similar conditions (126). In simulated gastrointestinal fluid, both *bsh* and *asp23* were upregulated, suggesting a coordinated stress response that enhances survival and colonization potential, supporting the probiotic potential of GX17. Additionally, the UspA (*SH1215*) was upregulated under multiple stress conditions, consistent with its role in *E. coli*, where it is induced by heat shock, oxidative stress, and carbon starvation and is essential for survival under growth-inhibiting conditions (127).

In the context of gene family analysis, comparative genomics plays an important role in identifying unique genes in specific strains. In this study, comparative genomics analysis of six strains of *L. plantarum* was performed, and it was found that the $Na^+/H^+$ antiporter-related gene (chr_2428) was unique to GX17, which plays an important role in maintaining the pH and $Na^+$ homeostasis of the cell. At the same time, by comparing the chromosome genomes of the six *L. plantarum* strains, it was found that the degree of conservation among the strains was different, but the *L. plantarum* GX17 used in our study did not differ much from the other five strains of *L. plantarum* with strong stress resistance in terms of genome structure, and except for the gene (chr_2428), the other stress resistance genes can also be found in the other strains. Combined with phenotypic experiments, it shows that *L. plantarum* GX17 has strong stress resistance and can be used as a high-quality candidate probiotic strain in production.

## Conclusions

In this study, a thorough analysis of the entire genome of *L. plantarum* GX17 uncovered numerous coding genes related to metabolism, including those involved in sugar, amino acid, and nucleotide metabolism. A comparison with the CAZy database revealed the strain's proficient carbohydrate utilization capabilities. In addition, the anti-stress ability of *L. plantarum* GX17 *in vitro* was analyzed, and found that it has good resistance to high temperature, low temperature, acid, alkali, salt, artificial gastrointestinal fluid, and strong antioxidant ability. Analysis of the genome and key gene transcription levels, the stress resistance of the strain was verified at the molecular level. These genes contribute to the synthesis of bacterial cells or their adhesion, allowing the strain to better resist the gastrointestinal environment and colonize the intestine, thereby exerting its probiotic role. Comparative genomic analysis showed that the genome of *L. plantarum* GX17 reorganized and transferred during evolution, allowing *L. plantarum* GX17 to better resist adverse environmental impacts. This highlights the potential of *L. plantarum* GX17 as a probiotic strain. This study provides a detailed view of the phenotype and genomic diversity of *L. plantarum* and helps to better understand the niche adaptability and functionality of organisms.

## ACKNOWLEDGMENTS

This research was supported by the Guangxi key research and development Program (AB241484045, AB23075145), the Laibing District Key Research and Development Program (240110, 240113), the Nanning Key Research and Development Program (NNKJ202408), the Fangchenggang District Key Research and Development Program (Fangke AB24002021), the Liangqing District Key Research and Development Program (202118), the Qingxiu key research and development Program (2022004), the Guangxi Academy of Agricultural Sciences stable funding research team project (2021YT109), the Guangxi Scientific Research Project (22-5, XKJ2325, XKJ2335), the Guangxi Innovation Team Construction Project of National Modern Agricultural Industry Technology System (nycytxgxcxtd-2021-09), the Special Project for the Construction of National beef cattle Industry Technical system (GZCYTX-03) and Qiankehe Platform Talent-YQK[2023]020. The funding bodies played no role in the design of the study and collection, analysis, interpretation of data, or writing of the manuscript.

Study design and planning: C.T.L., M.L., C.M., Y.Y., and C.L.L.; Data collection: Z.C., Jun L., Y.Y., and C.L.L.; Data analysis and statistics: H.B., L.T., L.W., Y.G., and Jing L.; Preparation of manuscript: Y.Y., C.L.L., and Z.P.; Review and editing: Z.Q., E.Z., and H.P. All authors read and approved the final manuscript.

## AUTHOR AFFILIATIONS

[1]Animal Science and Technology College, Guangxi University, Nanning, Guangxi, China
[2]Key Laboratory of Veterinary Biotechnology, Guangxi Veterinary Research Institute, Nanning, Guangxi, China
[3]Key Laboratory of China (Guangxi)-ASEAN Cross-border Animal Disease Prevention and Control, Ministry of Agriculture and Rural Affairs of China, Nanning, Guangxi, China
[4]Virginia Polytechnic Institute and State University, Blacksburg, Virginia, USA
[5]Guizhou Provincial Livestock and Poultry Genetic Resources Management Station, Guiyang, Guizhou, China
[6]Guangxi Academy of Agricultural Sciences, Nanning, Guangxi, China

## AUTHOR ORCIDs

Ezhen Zhang http://orcid.org/0009-0000-3388-121X
Hao Peng http://orcid.org/0000-0002-9619-5628

## FUNDING

| Funder | Grant(s) | Author(s) |
| --- | --- | --- |
| Guangxi Key Research and Development Program | AB23075145, AB241484045 | Yangyan Yin |
| | | Chunling Li |
| Special Project for the Construction of Nstional beef cattle Industry Technical system | GZCYTX-03 | Yu Gong |
| | | Jing Liu |
| | | Ezhen Zhang |
| | | Hao Peng |
| Qiankehe Platform Talent | YQK[2023]020 | Yu Gong |
| | | Jing Liu |
| | | Ezhen Zhang |
| | | Hao Peng |
| Laibing District Key Research and Development Program | 240110, 240113 | Yangyan Yin |
| | | Chunling Li |
| Naning Key Research and Development Program | NNKJ202408 | Zhe Pei |
| | | Zhongwei Chen |

| Funder | Grant(s) | Author(s) |
| --- | --- | --- |
|  |  | Huili Bai |
| Fangchenggang District Key Research and Development Program | Fangke AB24002021 | Huili Bai |
|  |  | Meiyi Lan |
| Liangqing District Key Research Key Research and Development Program | 202118 | Zhongwei Chen |
|  |  | Chunxia Ma |
| Qingxiu key research development Program | 2022004 | Jun Li |
|  |  | Ling Teng |
| Guangxi Academy of Agricultural Sciences | 2021YT109 | Yangyan Yin |
|  |  | Chunling Li |
|  |  | Leping Wang |
|  |  | Zhongsheng Qin |
|  |  | Hao Peng |
| Guangxi Scientific Research Project | 22-5, XKJ2325, XKJ2335 | Yangyan Yin |
|  |  | Chunling Li |
|  |  | Zhongwei Chen |
|  |  | Huili Bai |
|  |  | Ling Teng |
|  |  | Zhongsheng Qin |
|  |  | Hao Peng |
| Guangxi Innovation Team Construction Project of National Modern Agricultural Industry Technology System | nycytxgxcxtd-2021-09 | Zhe Pei |
|  |  | Huili Bai |
|  |  | Jun Li |
|  |  | Ling Teng |
|  |  | Ezhen Zhang |

## AUTHOR CONTRIBUTIONS

Yangyan Yin, Methodology, Writing – original draft, Writing – review and editing | Chunling Li, Conceptualization, Methodology, Writing – original draft, Writing – review and editing | Zhe Pei, Conceptualization, Supervision, Writing – review and editing | Changting Li, Investigation, Methodology, Resources, Supervision | Zhongwei Chen, Project administration, Resources, Supervision | Huili Bai, Data curation, Investigation, Supervision | Chunxia Ma, Data curation, Formal analysis, Supervision | Meiyi Lan, Conceptualization, Methodology, Resources | Jun Li, Project administration, Resources, Supervision | Yu Gong, Funding acquisition, Investigation, Resources, Supervision | Jing Liu, Funding acquisition, Investigation, Resources | Ling Teng, Conceptualization, Data curation, Investigation, Project administration | Leping Wang, Data curation, Software, Visualization | Zhongsheng Qin, Funding acquisition, Project administration, Resources | Ezhen Zhang, Conceptualization, Funding acquisition, Resources | Hao Peng, Conceptualization, Funding acquisition, Project administration, Resources, Writing – review and editing

## DATA AVAILABILITY

The data sets used and/or analyzed during the current study are available in the NCBI Sequence Read Archive repository [CP159198, CP159199, CP159200, CP159201, CP159202]. The GenBank accession number for accessing the *Lactiplantibacillus plantarum* GX17 genome sequence is [CP159198] (login URL: *Lactiplantibacillus plantarum* strain GX17 isolate feces chromosome, c - Nucleotide - NCBI) [CP159199] (login URL: *Lactiplantibacillus plantarum* strain GX17 isolate feces plasmid unnam - Nucleotide - NCBI) [CP159200] (login URL: *Lactiplantibacillus plantarum* strain GX17

isolate feces plasmid unnam - Nucleotide - NCBI) [CP159201] (login URL: *Lactiplantibacillus plantarum* strain GX17 isolate feces plasmid unnam - Nucleotide - NCBI) [CP159202] (login URL: *Lactiplantibacillus plantarum* strain GX17 isolate feces plasmid unnam - Nucleotide - NCBI).

## ADDITIONAL FILES

The following material is available online.

### Open Peer Review

**PEER REVIEW HISTORY (review-history.pdf).** An accounting of the reviewer comments and feedback.

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
