## [Reviewer comments · Microbiology Spectrum]

Microbiology Spectrum

Genomic and Stress Resistance Characterization of *Lactiplantibacillus plantarum* GX17, a Potential Probiotic for Animal Feed Applications

yangyan yin, Chunling Li, Zhe Pei, Changting Li, Zhongwei Chen, Huili Bai, Chunxia Ma, Meiyi Lan, Jun Li, Yu Gong, Jing Liu, Ling Teng, Leping Wang, Zhongsheng Qin, Ezhen Zhang, and Hao Peng

Corresponding Author(s): Hao Peng, Guangxi

Review Timeline:

Submission Date:	May 16, 2025
Editorial Decision:	June 11, 2025
Revision Received:	July 30, 2025
Accepted:	August 1, 2025

Editor: Benjamin Liu

Reviewer(s): The reviewers have opted to remain anonymous.

Transaction Report:

DOI: <https://doi.org/10.1128/spectrum.01243-25>

Re: Spectrum01243-25 (**Comprehensive Genome-wide Genomic Analysis and Stress Resistance Profiling of the Probiotic Strain *Lactiplantibacillus plantarum* GX17**)

Dear Prof. Hao Peng:

Thank you for the privilege of reviewing your work. Below you will find my comments, instructions from the Spectrum editorial office, and the reviewer comments.

Editor's comments:

Line 62-63: the authors introduced "enteric pathogens, including *Salmonella typhimurium*, *Escherichia coli*, and *Staphylococcus aureus*". However, *Staphylococcus aureus* is not an enteric pathogen. The authors should correct this by using "foodborne pathogens" rather than "enteric pathogens". Moreover, there are no references cited in this statement. More references on foodborne pathogens and food safety should be cited, with this one (Liu, B.M. History of global food safety, foodborne illness, and risk assessment. In History of Food and Nutrition Toxicology; Academic Press: Washington, DC, USA, 2023; pp. 301-316.) as an example (citing is optional).

Please return the manuscript within 30 days; if you cannot complete the modification within this time period, please contact me. If you do not wish to modify the manuscript and prefer to submit it to another journal, notify me immediately so that the manuscript may be formally withdrawn from consideration by Spectrum.

Revision Guidelines

Sincerely,
Benjamin Liu
Editor

Reviewer's comments for the authors:

Overall, this study provides analysis of the genomic and phenotypic features of a *L. plantarum* strain, which has been used as a probiotic supplement for yellow-feathered broilers. The data contributes to understanding of the niche adaptability and functional properties of the bacterial strain. The novelty of the study should be highlighted, with reference to a recent publication on the probiotic characteristics of *L. plantarum* (Zhang, Z., Niu, H., Qu, Q., Guo, D., Wan, X., Yang, Q., ... Liu, Z. (2025).

Advancements in *Lactiplantibacillus plantarum*: probiotic characteristics, gene editing technologies and applications. *Critical Reviews in Food Science and Nutrition*, 1-22. <https://doi.org/10.1080/10408398.2024.2448562>). The manuscript requires improvement in terms of clarity.

Title: Suggest to specify the application of the probiotic strain characterized in this study (for poultry or ..)

Abstract:

Line 30-32: the source and the identity of the bacterial strain were not clearly stated in the abstract.

Line 35-36: *Lactiplantibacillus plantarum* (*L.plantarum*) GX17 phenotype (or strain?) or genotype, this sentence should be revised.

Introduction:

References 30, 31, 32 are published studies on other bacteria. Are these studies relevant to the genomic analyses in this study?

Results and Discussion: Suggest to include comparative genomic analyses of other published *L. plantarum* strains (including reference strain). Highlight distinct phenotypic and genotypic features about GX17 that make it favourable as a probiotic strain.

Suggest to include transcriptomic validation of some key genes under stress or metabolic analyses to showcase the strain's application for probiotic use

Overall, this study provides analysis of the genomic and phenotypic features of a *L. plantarum* strain, which has been used as a probiotic supplement for yellow-feathered broilers. The data contributes to understanding of the niche adaptability and functional properties of the bacterial strain. The novelty of the study should be highlighted, with reference to a recent publication on the probiotic characteristics of *L. plantarum* (Zhang, Z., Niu, H., Qu, Q., Guo, D., Wan, X., Yang, Q., ... Liu, Z. (2025). *Advancements in Lactiplantibacillus plantarum: probiotic characteristics, gene editing technologies and applications. Critical Reviews in Food Science and Nutrition*, 1–22. <https://doi.org/10.1080/10408398.2024.2448562>). The manuscript requires improvement in terms of clarity.

Title: Suggest to specify the application of the probiotic strain characterized in this study (for poultry or ..)

Abstract:

Line 30-32: the source and the identity of the bacterial strain were not clearly stated in the abstract.

Line 35-36: *Lactiplantibacillus plantarum* (*L. plantarum*) GX17 phenotype (or strain?) or genotype, this sentence should be revised.

Introduction:

References 30, 31, 32 are published studies on other bacteria. Are these studies relevant to the genomic analyses in this study?

Results and Discussion: Suggest to include comparative genomic analyses of other published *L. plantarum* strains (including reference strain). Highlight distinct phenotypic and genotypic features about GX17 that make it favourable as a probiotic strain.

Suggest to include transcriptomic validation of some key genes under stress or metabolic analyses to showcase the strain's application for probiotic use

Dear Editor and Reviewers:

Thank you for your letter and for the reviewer's comments concerning our manuscript entitled "Comprehensive Genome-wide Genomic Analysis and Stress Resistance Profiling of the Probiotic Strain *Lactiplantibacillus plantarum* GX17" (Spectrum01243-25). Those comments are all valuable and very helpful for revising and improving our paper, as well as the important guiding significance of our researches. We have studied comments carefully and have made correction which we hope meet with approval. Your response to the general comments from the editor revised portion are marked in the yellow highlight in the "Marked-Up Manuscript" file. The main corrections in the paper and the responds to the editor and reviewer's comments are as follows:

Editor:

1. Line 62-63: the authors introduced "enteric pathogens, including *Salmonella typhimurium*, *Escherichia coli*, and *Staphylococcus aureus*". However, *Staphylococcus aureus* is not an enteric pathogen. The authors should correct this by using "foodborne pathogens" rather than "enteric pathogens". Moreover, there are no references cited in this statement. More references on foodborne pathogens and food safety should be cited, with this one (Liu, B.M. History of global food safety, foodborne illness, and risk assessment. In History of Food and Nutrition Toxicology; Academic Press: Washington, DC, USA, 2023; pp. 301-316.) as an example (citing is optional).

Response: Thank you very much for your valuable comments and suggestions. We have carefully revised the manuscript according to your recommendations. Specifically, we have corrected the term "enteric pathogens" to "foodborne pathogens" (Lines 63-65), as *Staphylococcus aureus* is indeed not an enteric pathogen. Additionally, we have added relevant references regarding foodborne pathogens and food safety, including the suggested reference (Liu, B.M., 2023). We appreciate your guidance, which has helped us improve the accuracy and quality of our manuscript.

Reviewer:

1. Title: Suggest to specify the application of the probiotic strain characterized in this study (for poultry or ..).

Response: In response to your comment, we have revised the manuscript's title to better reflect the intended application of the probiotic strain studied. The new title is: Genomic and Stress Resistance Characterization of *Lactiplantibacillus plantarum* GX17, a Potential Probiotic for Animal Feed Applications

This revision clarifies that the strain is being evaluated for its potential use in animal feed, which aligns with the scope and findings of our study.

2. Line 30-32: the source and the identity of the bacterial strain were not clearly stated in the abstract.

Response: In response to your suggestion, we have revised the abstract to clearly

state the source and identity of the bacterial strain. The updated abstract now includes the following information:

"This study comprehensively analyzed the genotypic and phenotypic characteristics of *Lactiplantibacillus plantarum* (*L. plantarum*) strains isolated from the intestines of healthy chicks, and assessed their potential as probiotics."(Lines 29–31)

This revision explicitly indicates that the *L. plantarum* GX17 was isolated from healthy chicks, thereby providing clear information on its source and origin.

3. Line 35-36: *Lactiplantibacillus plantarum* (*L.plantarum*) GX17 phenotype (or strain?) or genotype, this sentence should be revised.

Response: As the original sentence was potentially ambiguous and open to misinterpretation, we have carefully revised it according to your suggestion to ensure clarity and accuracy. The revised sentence is now presented in lines 37–40 of the revised manuscript.

4. Introduction: References 30, 31, 32 are published studies on other bacteria. Are these studies relevant to the genomic analyses in this study?

Response: Thank you for your insightful comment regarding the relevance of References 30, 31, and 32 in the Introduction section. You are absolutely right to point out that these references, which focus on genomic analyses of other bacteria, are not directly relevant to the genomic analyses presented in our study. In our initial draft, we included these references with the intention of providing a broader perspective on the application of genomic analyses in microbial research. However, upon re-evaluation, we realized that these studies do not align closely with the specific context of our work. Consequently, we have decided to remove the citations of References 30, 31, and 32 from the manuscript. Instead, we have added a new reference (Zhang, Z., et al., 2025) that is more pertinent to the genomic analysis of *L.plantarum*. The revised sentence now reads:

"With the rapid development of genomics and bioinformatics technology, genomic sequencing and analysis of potential probiotic strains, such as *L.plantarum*, have become extremely useful for obtaining sufficient information on safety and functional characteristics. This progress not only facilitates the understanding of their genetic background and physiological functions but also provides a scientific basis for strategies related to disease prevention and treatment."(Lines 73–77)

We believe that this revision enhances the relevance and accuracy of the references cited in our Introduction, ensuring that our study is properly contextualized within the existing literature.

5. Results and Discussion: Suggest to include comparative genomic analyses of other published *L. plantarum* strains (including reference strain). Highlight distinct phenotypic and genotypic features about GX17 that make it favorable as a probiotic strain.

Response: Thank you for your valuable suggestion regarding the inclusion of

comparative genomic analyses with other published *Lactobacillus plantarum* strains. We fully agree with your recommendation and have conducted comprehensive comparative genomic analyses between strain GX17 and other publicly available *L. plantarum* strains, including the reference strain. These analyses have allowed us to identify distinct phenotypic and genotypic features of GX17 that contribute to its probiotic potential. The results of these comparative analyses are presented in detail in the "Results"(Lines 290–320) section of our revised manuscript. Furthermore, in the "Discussion"(Lines 570–579) section, we elaborate on how these unique characteristics may confer advantages in probiotic applications. We believe that these additions significantly strengthen our study by providing genomic evidence for the probiotic properties of GX17 and highlighting its unique features compared to other *L. plantarum* strains. Thank you again for your insightful feedback, which has helped us improve the quality and impact of our work.

6. Suggest to include transcriptomic validation of some key genes under stress or metabolic analyses to showcase the strain's application for probiotic use.

Response: Thank you for your insightful suggestion. In response to your recommendation, we have incorporated transcriptomic validation of several key stress-related genes under various stress conditions to further demonstrate the probiotic potential of the GX17 strain. These additional analyses provide valuable insights into the molecular mechanisms underlying the strain's stress resistance, which is critical for its functionality as a probiotic. The results of the transcriptomic validation are presented in the "Results"(Lines 377–396) section, and their implications are discussed in detail in the "Discussion"(Lines 551–569) section of our revised manuscript. We believe that these new data significantly strengthen our study by providing experimental evidence to support the genomic predictions and highlighting the practical applications of GX17 as a probiotic.

Once again, we sincerely thank you for your constructive feedback and for the opportunity to improve our manuscript. We hope that the revisions adequately address your concerns and that the manuscript is now suitable for publication.

Re: Spectrum01243-25R1 (Genomic and Stress Resistance Characterization of *Lactiplantibacillus plantarum* GX17, a Potential Probiotic for Animal Feed Applications)

Dear Prof. Hao Peng:

Your manuscript has been accepted, and I am forwarding it to the ASM production staff for publication. Your paper will first be checked to make sure all elements meet the technical requirements. ASM staff will contact you if anything needs to be revised before copyediting and production can begin. Otherwise, you will be notified when your proofs are ready to be viewed.

Sincerely,
Benjamin Liu
Editor
Microbiology Spectrum